# Toward Memory-Aided World Models: Benchmarking via Spatial Consistency

## Abstract

The ability to simulate the world in a spatially consistent manner is a crucial requirements for effective world models. Such a model enables high-quality visual generation, and also ensures the reliability of world models for downstream tasks such as simulation and planning. Designing a memory module is a crucial component for addressing spatial consistency: such a model must not only retain long-horizon observational information, but also enables the construction of explicit or implicit internal spatial representations. However, there are no dataset designed to promote the development of memory modules by explicitly enforcing spatial consistency constraints. Furthermore, most existing benchmarks primarily emphasize visual coherence or generation quality, neglecting the requirement of long-range spatial consistency. To bridge this gap, we construct a dataset and corresponding benchmark by sampling 150 distinct locations within the open-world environment of Minecraft, collecting about 250 hours (20 million frames) of loop-based navigation videos with actions. Our dataset follows a curriculum design of sequence lengths, allowing models to learn spatial consistency on increasingly complex navigation trajectories. Furthermore, our data collection pipeline is easily extensible to new Minecraft environments and modules. Four representative world model baselines are evaluated on our benchmark. Dataset, benchmark, and code are open-sourced to support future research.

## 1 Introduction

Although recent advances in world models have significantly improved the visual quality of generated observations (Zhu et al., 2024; Ding et al., 2024; Brooks et al., 2024), these models often struggle to preserve spatial consistency over extended rollouts (Decart et al., 2024; Guo et al., 2025). Spatial consistency, the ability to preserve coherent and stable spatial structures across time, is essential for the reliability of world models in downstream applications such as model-based reinforcement learning (Bar et al., 2024; Alonso et al., 2024; Valevski et al., 2024), autonomous driving (Wang et al., 2023a; Gao et al., 2024), and model predictive control (Du et al., 2023; Hafner et al., 2020; 2024; Yang et al., 2023; 2024c). When violated, it can lead to severe failure modes: hallucinated structures, contradictions with prior observations, and visually incoherent scenes. For instance, in a world model guided navigation task, inconsistent rendering of revisited locations impedes global planning and loop closure.

The memory module plays a critical role in addressing spatial inconsistency. Due to the high dimensionality of image data, mainstream Transformer-based networks are typically limited to memorizing only a few or a dozen past frames. This is because the attention mechanism (Vaswani et al., 2017) incurs quadratic growth in computational and memory costs with respect to the context length, which quickly makes training and inference prohibitively expensive. However, real-world exploration trajectories often span hundreds or thousands of frames, and reconstructing an observation at a specific location may require access to information from hundreds of steps earlier. This makes it impractical to encode all historical observations directly into the model's context. To overcome this limitation, memory modules are typically designed as independent components that interact with the main model through read and write mechanisms (Graves et al., 2014), enabling long-term information retention. Additionally, a well-designed memory module should possess the capability to model spatial structures. This modeling can be explicit, such as through maps (Gupta et al., 2019; Yang et al., 2025), coordinate systems (Wang et al., 2023b), or graph-based representations that encode spatial

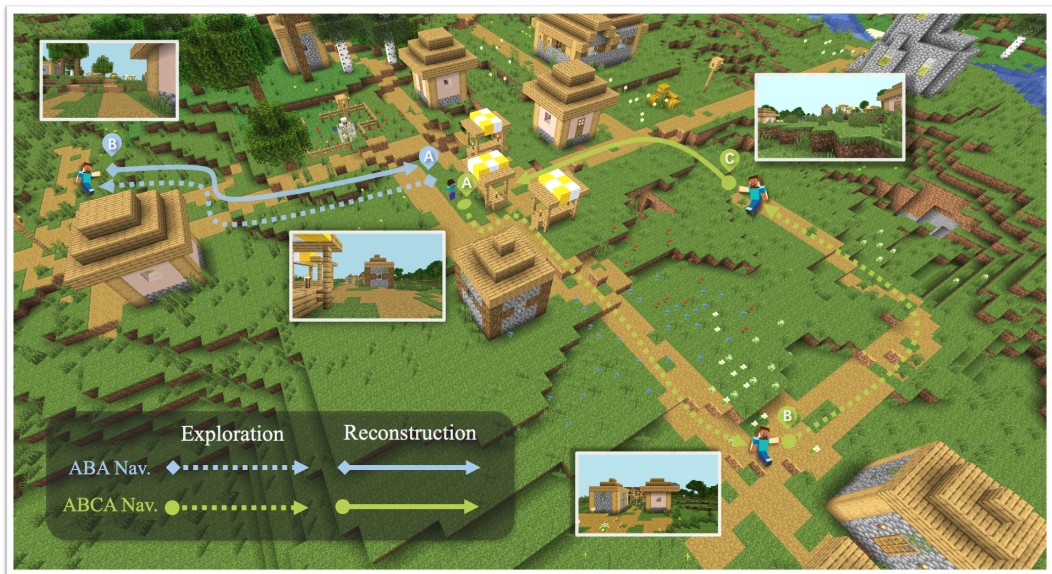

Figure 1: **The Loop-Style Trajectory Data for Training and Benchmarking**. To be able to explicitly enforce spatial consistency, our dataset follows a loop-style navigation trajectory, including both ABA and ABCA types. Our benchmark design explicitly separates the generation phase from the exploration phase and only evaluates the video quality of the generation part.

relationships directly (Savinov et al., 2018; Kim et al., 2022). Alternatively, it can be implicit, using high-dimensional embeddings or attention mechanisms to capture relative spatial relationships and similarity across observations (Parisotto & Salakhutdinov, 2017; Wang* et al., 2025). Regardless of the form, the central goal of spatial modeling within memory is to maintain a coherent representation of space over time.

Although memory modules are essential for enabling world models to maintain spatial consistency, their design, training, and evaluation remain underexplored. A major reason for this gap is the absence of datasets that explicitly demand spatial consistency. Most existing datasets are designed for open-ended exploration, where agents continually encounter novel structures and objects while rarely revisiting previously seen locations. As a result, world models trained on such data tend to rely on global environmental priors to hallucinate plausible next frames, rather than leveraging episodic memory to reconstruct previously observed content. Meanwhile, evaluation metrics often prioritize visual fidelity and short-term temporal smoothness over long-term spatial coherence or logical consistency. This leads to models that function more as visually impressive but impractical "dream machines", lacking reliability for downstream decision-making tasks.

Therefore, we argue that the dataset must feature looped trajectories that revisit the same locations and objects from diverse viewpoints. We construct such a dataset, LOOPNAV, in the open-world environment of Minecraft, chosen for its rich environmental diversity, extensive community support, and growing research interest (Baker et al., 2022; Cai et al., 2023; 2025b; Lifshitz et al., 2024; Decart et al., 2024). The dataset comprises over 250 hours of **loop-style** navigation trajectories across 147 diverse locations. Each trajectory follows a loop exploration pattern (e.g., $A \rightarrow B \rightarrow A$ or $A \rightarrow B \rightarrow C \rightarrow A$, where $A, B, C$ denote different locations), ensuring that the same locations and layouts are revisited from varying camera views and times. This loop-based data structure naturally incentivizes models to learn long-horizon spatial consistency. Besides, we carefully select spawning positions (e.g., villages) with diverse landmark objects to ensure that observations from different locations are visually distinguishable. To support progressive learning, the dataset also includes a curriculum of sequence lengths, enabling models to transition from short-term to long-term tasks. As for the benchmarking, we propose a **explore-then-generate** approach to evaluate the spatial consistency of world models. Specifically, we sample a closed-loop trajectory (e.g., $A \rightarrow B \rightarrow A$) from the test set, where the first half (e.g., $A \rightarrow B$) serves as the context input to the world model, and the second half (e.g., $B \rightarrow A$) is used as the generation target. By evaluating four representative

baseline world models, we find that their performance on spatial consistency remains far from satisfactory, highlighting the urgent need for dedicated datasets and advanced model architectures.

## 2 RELATED WORK

### 2.1 WORLD MODELS

The goal of a world model is to simulate the environment: Given the current state and action, it predicts the next state and the corresponding reward. World models were originally proposed to improve sample efficiency in reinforcement learning(Oh et al., 2015; Ha & Schmidhuber, 2018). In the context of model-based reinforcement learning, numerous studies have explored various architectural designs of world models. Dreamer (Hafner et al., 2020; 2022; 2024) uses a Recurrent State Space Model (RSSM) and achieves human-level performance on Atari games. TWM (Robine et al., 2023) adapts DreamerV2's RSSM to use a transformer architecture. IRIS (Micheli et al., 2023) builds image tokens with discrete autoencoder and adopts an autoregressive transformer. DIAMOND (Alonso et al., 2024) leverages diffusion models to generate future frames. These architectures, to varying degrees, retain information from the past to aid in future image prediction; however, they still lack an effective memory design capable of maintaining long-horizon spatial consistency.

Beyond model-based reinforcement learning, the potential of world models has also been increasingly explored in other domains. Playable Video Generation (Menapace et al., 2021), gameNGen (Valevski et al., 2024), Oasis (Decart et al., 2024) investigate the potential of using world models as game engines. GAIA (Hu et al., 2023), DriveDreamer Wang et al. (2023a), vista (Gao et al., 2024) explores their application in autonomous driving. In robotics, two of the most significant challenges are the scarcity of training data and the high cost of model training. The emergence of world models offers a promising direction for addressing these two major challenges. UniSim (Yang et al., 2023) learns a universal simulator for robot manipulation. DayDreamer (Wu et al., 2022) employs world model for real-world robot learning. NWM (Bar et al., 2024) investigates visual navigation problem in world model. In almost all downstream tasks—especially those involving decision-making such as autonomous driving and navigation—maintaining spatial consistency is a critical requirement.

From a more unified perspective, world models can be viewed as action-conditioned video generation models. In computer vision, generating videos has been a long standing challenge (Yang et al., 2024c). Recent approaches leverage transformers , and diffusion models (Chen et al., 2024; Song et al., 2025) to generate longer, more coherent video sequences. From the perspective of video modeling, maintaining spatial consistency across frames—especially over long temporal horizons—remains a critical and unsolved challenge.

### 2.2 MINECRAFT AS AN AI TESTBED

Minecraft is an open-world environment characterized by diverse terrains and rich interaction dynamics. As one of the most popular games globally, it benefits from extensive community-driven resources and an active user base. Recent research has increasingly adopted Minecraft as a platform for training generative agents (Baker et al., 2022; Lifshitz et al., 2024; Cai et al., 2023; 2024; 2025b;a; Zhao et al., 2024) and constructing digital world models (Guo et al., 2025; Decart et al., 2024; Hong et al., 2024; Song et al., 2025). Several Minecraft-based datasets and benchmarks have been introduced, including MineDoJo (Fan et al., 2022), which provides multimodal knowledge and human gameplay videos sourced from YouTube; VPT (Baker et al., 2022), which releases large-scale trajectories of human players engaged in building and mining; and MineRL (Guss et al., 2019), which focuses on trajectories related to cave exploration. However, these efforts primarily emphasize prospective exploration and object interaction, with limited attention to the role of historical context. In contrast, our proposed dataset and benchmark are designed to promote and assess models' abilities to leverage past observations and perform spatial reasoning. Our data collection pipeline is built on top of Mineflayer (Contributors, 2013–2024), a widely used rule-based Minecraft automation framework.

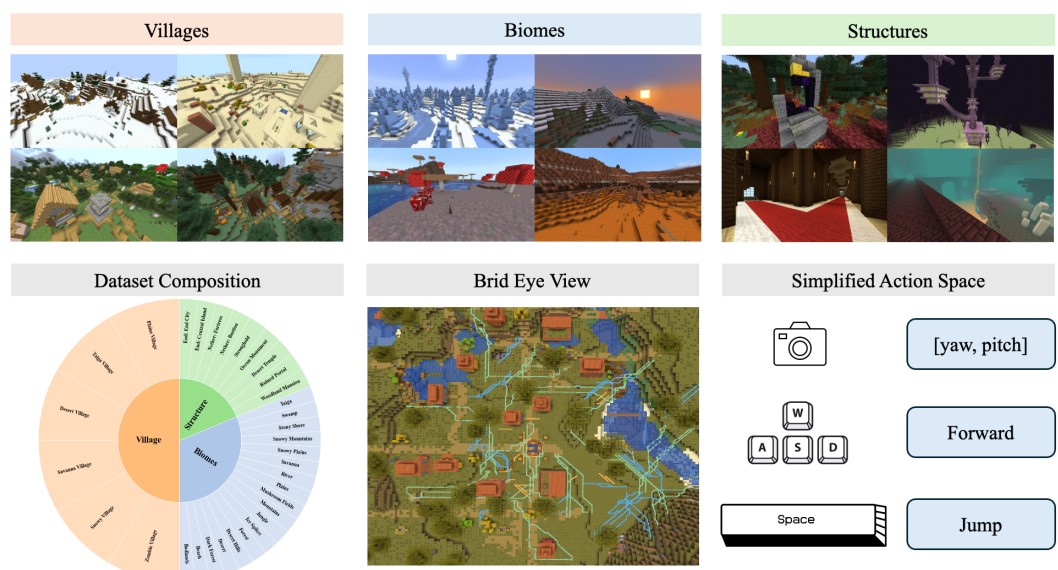

Figure 2: **Overview of the Minecraft Elements**. Top row, left to right: Examples of villages, biomes, and structures in Minecraft. Bottom row, left to right: composition of sampling locations in our dataset; a bird-eye view of real ABA-type exploration trajectories (display steps 5, 15, and 30 for simplicity); Simplified Action Space used for data collection and agent interaction.

# 3 LoopNav Dataset

## 3.1 Environment

We collect data in the open-world environment of Minecraft, selected for the following reasons:

- **Rich environmental diversity**: Minecraft worlds are procedurally generated using random seeds, featuring a wide range of biomes and uniquely structured villages, enabling diverse and non-repetitive environments.

- **Extensive community resources**: Minecraft has a mature modding ecosystem that allows us to incorporate a wide variety of custom elements and tasks. In addition, there is a large amount of publicly available Minecraft gameplay footage on platforms like YouTube, which could be leveraged for future large-scale pretraining.

- **Rising research interest**: An increasing number of recent works have begun to study world modeling and agent learning in Minecraft, making it a timely and relevant platform for evaluating spatial consistency and memory mechanisms.

- **Efficient simulation**: Compared to real-world data collection, Minecraft enables faster, cheaper, and highly parallelizable simulation, making it well-suited for large-scale controlled data generation.

## 3.2 Collection Pipeline

We propose three principles for trajectory data collection: *visual discriminability*, *loop closure*, and *curriculum-based progression*. Below, we detail each principle, followed by our implementation.

> **Principle 1: Visual Discriminability**
>
> *The sequence of observations along the trajectory should be visually distinguishable over time to capture meaningful scene variation.*

Within this procedurally generated world, we identify 6 categories that encompass a total of 120 distinct villages, offering a wide range of terrain and architectural layouts. As players explore a village, visual elements such as houses and farmlands, distinguished by their shapes and spatial arrangements, serve as salient landmarks that help construct a cognitive map of the environment. In addition to the villages, we also collect trajectories from 18 different biome types and 8 unique Minecraft-specific structures (e.g., desert temples, ruined portals), resulting in 146 distinct spatial locations in total. Examples of villages, biomes, and structures are shown in Figure 2, and detailed descriptions can be found in Appendix B. This diverse selection ensures variation in both visual appearance and spatial layout, which is critical for evaluating long-horizon spatial consistency.

> **Principle 2: Loop Closure**
>
> *The generated trajectories must form at least one spatial loop, ensuring repeated visits to the same locations.*

We propose two types of trajectory structures: $A \to B \to A$ and $A \to B \to C \to A$, where A, B, and C denote distinct locations within the environment. Without loss of generality, we use the $A \to B \to A$ trajectory as a representative case to illustrate our data collection process. As detailed in Algorithm 1, we first select a location $S$ from one of our collected sites (visually distinguishable). Then, we randomly sample a starting point $A$ within the start range of $S$, and teleport the agent to $A$. This is done to ensure diversity in the starting positions within the same location. Start range was set to 20 blocks in practice. Once at location $A$, a target location $B$ is sampled within a specified navigation range. At this location, trajectory recording begins: the agent performs a full 360-degree rotation at location $A$ to observe the surrounding environment and establish spatial context. It then navigates from $A$ to $B$ using the A* algorithm, and subsequently returns from $B$ to $A$. The recording ends upon the agent's return to location $A$, completing the loop. We put visualization of our dataset, including frames and bird-eye view in Appendix B.4.

---

**Algorithm 1** $A \to B \to A$ Navigation Data Collection in Location $S$

---

**Require:** Start Range $R_{st}$, Navigation Range $R_{nav}$, location $S$
**Ensure:** A navigation trajectory $T$ from point $A$ to point $B$ and back to $A$
1: $A \leftarrow \text{SAMPLEPOINT}(S, R_{st})$
2: $B \leftarrow \text{SAMPLEPOINT}(A, R_{nav})$
3: Trajectory $T \leftarrow []$
4: Teleport the agent to location $A$
5: Agent performs a 360° rotation at point $A$ to observe surroundings
6: $T \leftarrow \text{NAVIGATE}(A, B, T)$
7: $T \leftarrow \text{NAVIGATE}(B, A, T)$
8: **return** Trajectory $T$

---

```
1: function SAMPLEPOINT(A = (x, y), r)        1: function NAVIGATE(A, B, T)
2:     while True do                           2:     S ← A
3:         Sample Δx ∼ Uniform(−r, r)          3:     while S ≠ B do
4:         Sample Δy ∼ Uniform(−r, r)          4:         P ← ASTARPLAN(S, B)
5:         d ← √((Δx)² + (Δy)²)                5:         for all P_i ∈ P do
6:         if d < 0.8 · r then                 6:             a ← GETACTION(S, P_i)
7:             continue                        7:             S ← PERFORMACTION(S, a)
8:         end if                              8:             Append (S, a) to T
9:         B ← (x + Δx, y + Δy)                9:         end for
10:        if ISVALIDBLOCK(B) then            10:     end while
11:            return B                        11:     return T
12:        end if                             12: end function
13:    end while
14: end function
```

---

> **Principle 3: Curriculum-based Progression**
>
> *The trajectories follow a curriculum design, gradually expanding the spatial exploration radius to support progressive learning and evaluation.*

According to the exploration radius, we define four difficulty levels: 5, 15, 30, and 50 blocks. To improve training robustness and introduce variability, the target location $B$ is not sampled at a fixed distance. Instead, for a difficulty level $x$, we sample distances within the interval $[0.8x, \sqrt{2}x]$, ensuring diverse trajectory lengths while minimizing overlap between different difficulty levels. The full sampling procedure is also detailed in Algorithm 1.

We collect 20 trajectories for each combination of location and exploration radius. The final dataset contains approximately 20 million frames, amounting to roughly 250 hours of navigation trajectories.

### 3.3 IMPLEMENTATION DETAILS

We set up a local Minecraft server and use the Mineflayer platform to control the agent and collect trajectory data. For path planning, we employ the Mineflayer Pathfinder[1] plugin, which computes shortest paths between waypoints using the A* algorithm. We use the Prismarine Viewer[2] to render and visualize the agent's behavior and collected observations. To better support learning from the collected data and simulate human-like control dynamics, we make several key modifications to the default behavior of these plugins. These choices define some of the core characteristics of our dataset:

**Restricted action space**: The agent is limited to using only three actions during navigation — forward, jump, and camera rotation — closely matching the action primitives of a human player in Minecraft.

**Sequential action execution**: At any given time step, only a single action is allowed. That is, the agent cannot rotate the camera and move forward simultaneously. This avoids entangled motion patterns and improves the clarity of spatial transitions in the data.

**Smooth camera control**: We impose a maximum angular velocity on camera rotations to prevent abrupt viewpoint changes between frames, ensuring more stable visual continuity for training.

**Removal of irrelevant elements**: To focus the dataset on pure navigation, we disabled all entity spawning (e.g., mobs), and removed UI elements such as the hotbar and hands, eliminating visual distractions and ensuring task purity.

We introduce the details of these plugins and our modification in Appendix A.

For each recorded frame, we store the following information $(S_t, A_t, P_t, C_t, G)$:

- The current RGB image observation $S_t$ rendered from the agent's first-person view.
- The action executed at this timestep $A_t = (\text{forward}, \text{jump}, \Delta\text{yaw}, \Delta\text{pitch})$.
- The agent's current position $P_t = (x_t, y_t, z_t)$.
- The camera orientation $C_t = (\text{yaw}, \text{pitch})$.
- The current navigation target's coordinates $G = (x_G, y_G, z_G)$.

The format of the collected data are detailed in the Appendix B.2. Importantly, the recorded action corresponds to the action executed at time $t$, i.e., $A_t$. Its effects are reflected in the next state $S_{t+1}$ and beyond. Data is recorded at 20 Hz, meaning a time interval of 0.05 seconds between two consecutive frames. This frequency aligns with the physical tick rate of Minecraft, ensuring consistent synchronization between control and simulation.

## 4 BENCHMARK

Our benchmark is designed around $A \to B \to A$ loop trajectories. In this setting, we treat the $A \to B$ segment as an exploration phase, which serves as the contextual input to the model, and the

---

[1]https://github.com/PrismarineJS/mineflayer-pathfinder

[2]https://github.com/PrismarineJS/prismarine-viewer

$B \rightarrow A$ segment as a reconstruction phase, where the agent returns to a previously visited location. We provide an illustration of these phases in Figure 1. As such, we evaluate model performance only on the $B \rightarrow A$ segment, as it provides a clear test of the model's ability to maintain spatial consistency during long-horizon rollouts. We compute three widely used video generation metrics:

**Fréchet Video Distance (FVD)** We adopt Fréchet Video Distance (FVD) (Unterthiner et al., 2019) as a primary metric. FVD computes the distance between the distribution of real and generated video features extracted by an Inflated 3D ConvNet (I3D) (Carreira & Zisserman, 2018), capturing both spatial and temporal statistics at a high level.

**Learned Perceptual Image Patch Similarity (LPIPS)** We use LPIPS (Zhang et al., 2018) to evaluate the perceptual similarity between generated and ground truth frames. LPIPS leverages deep features from pretrained networks (e.g., VGG) to assess semantic-level differences, which aligns better with human judgment than pixel-wise metrics. It is robust to minor texture or color variations while remaining sensitive to object structure and layout.

**Structural Similarity Index Measure (SSIM)** SSIM is employed to complement LPIPS by quantifying structural fidelity and luminance similarity between generated and reference frames. While it is more sensitive to low-level details than LPIPS, it still emphasizes perceptual structure over exact pixel match.

In addition to the above quantitative metrics, we argue that **qualitative evaluation** plays an equally critical role in assessing world model performance. Human visual inspection remains the most straightforward and effective way to determine whether the model accurately reconstructs previously seen environments and preserves spatial consistency during long-horizon rollouts. This is especially important as no single quantitative metric fully captures the semantic fidelity or spatial coherence of the generated trajectories.

Similarly, for longer $A \rightarrow B \rightarrow C \rightarrow A$ trajectories, we evaluate only the $C \rightarrow A$ segment, treating $A \rightarrow B \rightarrow C$ as the exploration context. This evaluation strategy emphasizes the model's capacity to reconstruct previously visited locations after long-horizon exploration. It is worth noting that in $A \rightarrow B \rightarrow C \rightarrow A$ trajectories, the return path from $C$ to $A$ is not guaranteed to be fully covered by the earlier observations from exploration context $A \rightarrow B \rightarrow C$. This discrepancy can theoretically occur, particularly in open-world environments. However, in practice, we constrain the exploration range such that the majority of $C \rightarrow A$ trajectories traverse areas that have been previously observed, albeit from different viewpoints or at different times. As a result, the model typically has access to sufficient contextual information to support the reconstruction of the return path, making our evaluation of spatial consistency both practical and meaningful.

## 5 EXPERIMENTS

### 5.1 BASELINES

We evaluate the following baselines on our dataset and benchmark.

**Oasis** Oasis (Decart et al., 2024) is a world model with a ViT (Dosovitskiy et al., 2021) model as spatial autoencoder, and a DiT (Peebles & Xie, 2022) model as latent diffusion backbone. Oasis generates frames autoregressively, with the ability to condition each frame on user input. The model was trained using Diffusion Forcing (Chen et al., 2024). We use the publicly available `Oasis-500M` model, with the context length set to 32. Since the training code for Oasis is not publicly available, we directly evaluate the pretrained model provided by the authors. As the model is pretrained on Minecraft data, we consider this evaluation setting to be reasonable and representative for our task domain.

**Mineworld** Mineworld (Guo et al., 2025) is another interactive world model pretrained on Minecraft data. Different from oasis, MineWorld base on pure transformer structure. It is driven by a visual-action autoregressive Transformer, which takes paired game scenes and corresponding actions as input, and generates consequent new scenes following the actions. We also directly evaluated the largest available pretrained checkpoint `1200M-32f`. The context length of Mineworld is 32.

**DIAMOND** DIAMOND (Alonso et al., 2024) is a diffusion-based world model built upon the UNet architecture (Ronneberger et al., 2015). It generates video frames autoregressively, conditioning on

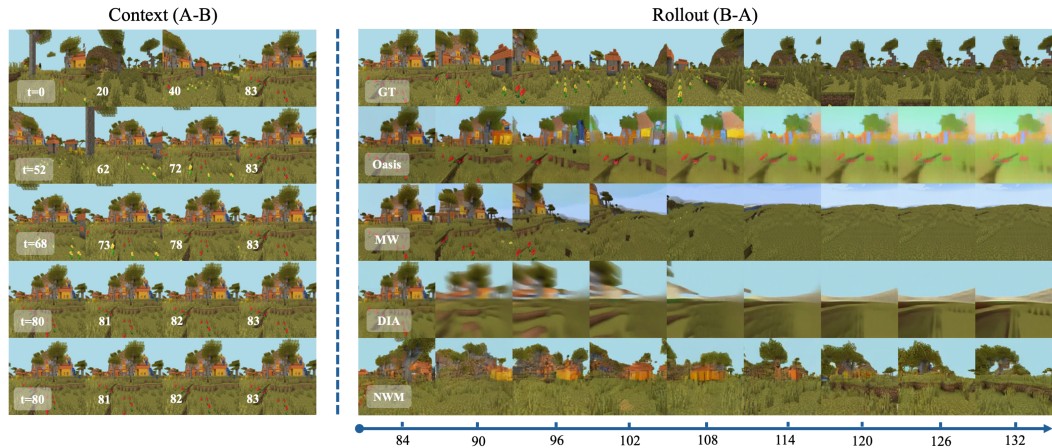

Figure 3: **Qualitative Result of Four Baselines**. Top to buttom: Ground Truth(GT), Oasis, Mineworld(MW), DIAMOND(DIA), Navigation World Model(NWM). Leftmost label "t=..." indicates the start context range accepted by each model. For example, for the Oasis model, "t=52" means frames 52 to 83 are used as context. All models begin rollout from frame 84.

both past observations and actions, allowing it to model complex temporal dynamics in sequential decision-making environments. In its original design, DIAMOND is used as part of a model-based reinforcement learning pipeline, where the learned world model is used to train an RL agent and evaluated on the Atari 100k benchmark. And in CS:GO branch, DIAMOND also exhibits extraordinary ability to modeling dynamics. In our experiments, we follow the default setting on CS:GO experiment of DIAMOND. We pretrain a model using our dataset from scratch. The context length set to 32, aligning with oasis and Mineworld.

**Navigation World Model** NWM (Bar et al., 2024) is a controllable video generation model that predicts future visual observations conditioned on past observations and navigation actions. It enables trajectory planning by simulating possible navigations and evaluating their success in reaching the goal. NWM introduces a novel Conditional Diffusion Transformer (CDiT) trained using a diffusion-based framework. Unlike aformention models, Navigation World Model did not use actions as condition, but use $(x, z)$ position and absolute yaw value as condition. We pretrained `CDiT-B/2` model from scratch with context length set to 4, due to limitation of computational resources.

The implementation details and additional results of the four baselines are provided in the Appendix C.

## 5.2 EXPERIMENT SETTINGS

Although our dataset includes data from villages, biomes, and structures, we focus solely on village environments for training and evaluation. This decision is based on the large number of villages in Minecraft and the inherent diversity of them. Our dataset includes six distinct village types, with 20 unique villages collected for each type. Villages 1–16 are used for training set, village 17 and 18 is used as the validation set, and village 19 and 20 serves as the test set. That is, the training set contains 15,360 trajectories (approximately 15 million frames), while the validation and test sets include 1920 trajectories (around 2 million frames). We provide the training and evaluation splits as part of our released dataset.

**Evaluation** We evaluate each model across all navigation ranges. Due to the large scale of the test dataset, we do not perform evaluation on the entire set. Instead, we sample the first three trajectories in lexicographic order from each of the 6 villages with index 20, resulting in 18 evaluation trajectories per navigation range. Each trajectory is assessed using the metrics described in the benchmark section. We report the average performance across all 18 trajectories, as shown in Table 1. In addition, we randomly select one trajectory to present qualitative results, shown in Figure 3.

## 5.3 ANALYSIS

**Overall Performance.** The results of all baseline models are far from satisfactory, none of the four models achieve strong performance. A key limitation lies in the absence of explicit memory modules across all baselines. Most models operate with a context length of 32 frames, whereas even the simplest evaluation tasks with a navigation range of 5 typically involve 60–70 frames. As such, the models understandably fail to demonstrate spatial consistency over longer sequences. However, we believe that our dataset can serve as a valuable resource for developing and evaluating more advanced models with improved memory mechanisms in the future.

**Model Collapse** A significant issue observed in both DIAMOND and OASIS is model collapse, especially as the prediction horizon increases(Figure 3). In these cases, the models gradually degenerate from minor imperfections to complete visual failure over time. This degradation is reflected in the high FVD score. The progressive nature of this collapse emphasizes the limitations of these models in maintaining stability over longer rollouts.

**Minor Differences Across Ranges** We find that, our result do not show a consistent decline with increasing navigation range. This suggests that the models are unable to effectively handle even the simplest case (navigation range of 5), and thus further increases in difficulty do not significantly impact the results.

Table 1: **Evaluation Results of Four Baselines under Different Navigation Ranges**. The ✓ icon indicates that the model was trained on our dataset, while ✗ indicates using a pretrained checkpoint.

| Model | Train | Context | SSIM ↑ | | LPIPS ↓ | | FVD ↓ | |
|---|---|---|---|---|---|---|---|---|
| | | | ABA | ABCA | ABA | ABCA | ABA | ABCA |
| Oasis-5 | ✗ | 32 | $0.36_{\pm 0.13}$ | $0.34_{\pm 0.12}$ | $0.76_{\pm 0.09}$ | $0.82_{\pm 0.11}$ | $2615_{\pm 1067}$ | $2583_{\pm 647}$ |
| Oasis-15 | ✗ | 32 | $0.37_{\pm 0.12}$ | $0.38_{\pm 0.14}$ | $0.82_{\pm 0.08}$ | $0.81_{\pm 0.10}$ | $2516_{\pm 567}$ | $3146_{\pm 1055}$ |
| Oasis-30 | ✗ | 32 | $0.33_{\pm 0.11}$ | $0.35_{\pm 0.11}$ | $0.86_{\pm 0.08}$ | $0.85_{\pm 0.09}$ | $3131_{\pm 713}$ | $3199_{\pm 1000}$ |
| Oasis-50 | ✗ | 32 | $0.36_{\pm 0.12}$ | $0.36_{\pm 0.11}$ | $0.86_{\pm 0.09}$ | $0.83_{\pm 0.07}$ | $3334_{\pm 658}$ | $3162_{\pm 1245}$ |
| Mineworld-5 | ✗ | 32 | $0.31_{\pm 0.09}$ | $0.32_{\pm 0.10}$ | $0.73_{\pm 0.05}$ | $0.72_{\pm 0.07}$ | $2089_{\pm 1007}$ | $1914_{\pm 660}$ |
| Mineworld-15 | ✗ | 32 | $0.34_{\pm 0.13}$ | $0.32_{\pm 0.11}$ | $0.74_{\pm 0.08}$ | $0.74_{\pm 0.07}$ | $2367_{\pm 770}$ | $2009_{\pm 921}$ |
| Mineworld-30 | ✗ | 32 | $0.33_{\pm 0.13}$ | $0.28_{\pm 0.09}$ | $0.77_{\pm 0.08}$ | $0.77_{\pm 0.08}$ | $2316_{\pm 945}$ | $2094_{\pm 1047}$ |
| Mineworld-50 | ✗ | 32 | $0.31_{\pm 0.16}$ | $0.32_{\pm 0.12}$ | $0.78_{\pm 0.12}$ | $0.75_{\pm 0.10}$ | $2077_{\pm 632}$ | $2144_{\pm 898}$ |
| DIAMOND-5 | ✓ | 32 | $0.40_{\pm 0.10}$ | $0.37_{\pm 0.09}$ | $0.75_{\pm 0.09}$ | $0.79_{\pm 0.09}$ | $3353_{\pm 1242}$ | $3336_{\pm 1392}$ |
| DIAMOND-15 | ✓ | 32 | $0.38_{\pm 0.10}$ | $0.39_{\pm 0.10}$ | $0.78_{\pm 0.08}$ | $0.79_{\pm 0.09}$ | $3691_{\pm 937}$ | $3302_{\pm 1191}$ |
| DIAMOND-30 | ✓ | 32 | $0.37_{\pm 0.10}$ | $0.35_{\pm 0.10}$ | $0.81_{\pm 0.07}$ | $0.81_{\pm 0.08}$ | $3708_{\pm 1243}$ | $3473_{\pm 1355}$ |
| DIAMOND-50 | ✓ | 32 | $0.37_{\pm 0.10}$ | $0.38_{\pm 0.09}$ | $0.83_{\pm 0.09}$ | $0.81_{\pm 0.08}$ | $3249_{\pm 833}$ | $2994_{\pm 906}$ |
| NWM-5 | ✓ | 4 | $0.33_{\pm 0.11}$ | $0.31_{\pm 0.09}$ | $0.64_{\pm 0.05}$ | $0.67_{\pm 0.05}$ | $1950_{\pm 380}$ | $2240_{\pm 664}$ |
| NWM-15 | ✓ | 4 | $0.30_{\pm 0.12}$ | $0.33_{\pm 0.12}$ | $0.67_{\pm 0.03}$ | $0.65_{\pm 0.05}$ | $2132_{\pm 916}$ | $2338_{\pm 1010}$ |
| NWM-30 | ✓ | 4 | $0.32_{\pm 0.11}$ | $0.30_{\pm 0.11}$ | $0.69_{\pm 0.04}$ | $0.71_{\pm 0.03}$ | $1893_{\pm 1047}$ | $2437_{\pm 429}$ |
| NWM-50 | ✓ | 4 | $0.28_{\pm 0.13}$ | $0.33_{\pm 0.11}$ | $0.72_{\pm 0.08}$ | $0.65_{\pm 0.04}$ | $2715_{\pm 883}$ | $1537_{\pm 415}$ |

## 6 LIMITATIONS

**Lack of depth data**: Many navigation and 3D modeling tasks require depth information to assist with memory modeling. While recent works like Depth Anything (Yang et al., 2024a;b) can generate synthetic depth data, there is still a gap between these synthetic depth maps and ground truth 3D data. We believe that real-time depth acquisition from the Minecraft server using scanning techniques could address this, but it would introduce substantial computational overhead.

**Static structures in the dataset**: The structures in our dataset (such as houses and terrain) are static, and we intentionally excluded moving objects like mobs. This decision was made to simplify the task of modeling spatial consistency. However, in real-world environments, the positions of vehicles, people, and other dynamic objects change over time. Maintaining spatial consistency in such dynamic environments is a more challenging task and a promising direction for future exploration. We leave this task for future work.

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

## APPENDIX OVERVIEW

Our appendix is organized as follows:

- **Appendix A** describes the platform setup for our data collection, including the MC environment setup and three javascript plugins we modified to manipulate the bot and collect data.
- **Appendix B** presents detials of our dataset, including B.2 data storage format, B.3 statistical information and B.4 visualization of our LOOPNAV dataset.
- **Appendix C** presents the detailed baseline experimental settings and additional experimental results:

    **C.1**: Open-oasis, **C.2**: Mineworld, **C.3**: DIAMOND, **C.4**: Navigation World Model.

## A  PLATFORM SETUP

### A.1  MINECRAFT

Minecraft is an open-world environment characterized by diverse terrains and rich interaction dynamics. We established a local Mineworld server running **Java Edition** version **1.16.5**, utilizing the seed value of **42**. The server was configured in survival mode with the difficulty set to peaceful, and mob spawning was disabled to eliminate the presence of animals. This configuration was implemented to prevent interference from mobs during trajectory-related experiments.

We utilized the Chunkbase website [3] to locate various villages and biomes. A total of 120 villages were collected—comprising 6 types, with 20 samples each—along with locations of 18 distinct biomes and 8 structures. The agent was teleported to each initial position using Minecraft's teleport command before commencing navigation.

### A.2  MINEFLAYER

Minefayer [4] is a powerful open-source JavaScript API designed for flexibly controlling in-game bots within Minecraft. It interacts with the Minecraft server by continuously reading block information around the bot based on its current x, y, and z coordinates, which is loaded locally. Users are allowed to specify actions for the bot to perform. Mineflayer simulates the bot's movement physically, computes its resulting state and position after each action, and subsequently sends updated coordinates (x, y, z), yaw, and pitch values to the server. Benefiting from this flexible interaction mechanism, diverse action policies can be developed to enable the agent to accomplish various tasks.

We introduced a modification to Mineflayer by constraining the horizontal rotation speed per time step, such that each action can result in a **maximum rotation of 0.1 radians**. This adjustment ensures smoother and more gradual viewpoint transitions.

### A.3  MINEFLAYER-PATHFINDER

The Mineflayer Pathfinder [5] is an application built on top of Mineflayer. It is capable of planning a path from the bot's current position to a target coordinate (x, y, z) using the A* algorithm. The bot is then controlled through a series of actions to navigate toward the destination.

---

[3]http://www.chunkbase.com/

[4]https://github.com/PrismarineJS/mineflayer

[5]https://github.com/PrismarineJS/mineflayer-pathfinder

In the original implementation, camera rotation and movement were coupled within the same action. In our modified version, at any given time step, **only a single action is allowed**. That is, the agent cannot rotate the camera and move forward simultaneously. This avoids entangled motion patterns and improves the clarity of spatial transitions in the data.

Furthermore, the original Mineflayer Pathfinder often exhibited sharp turns and other non-smooth behaviors during navigation, particularly causing rapid camera jitter near jagged block edges. To mitigate this issue, we prevent the bot from moving along the edges of 1×1 blocks, thereby eliminating such undesirable motion artifacts.

### A.4 PRISMARINE-VIEWER

Prismarine Viewer [6] is a complementary plugin designed to work alongside Mineflayer, capable of rendering the Minecraft environment from either a first-person or third-person perspective. In this sense, the combination of Mineflayer and Prismarine Viewer effectively constitutes a functional Minecraft client. We employed Prismarine Viewer to render and generate gameplay visuals.

The rendering logic of the viewer operates by periodically capturing the current coordinates (x, y, z) yaw pitch and rendering the scene at a fixed time interval. We set this sampling frequency to 50 Hz, matching both Minecraft's default refresh rate and the action-sending frequency. As a result, each frame is rendered every 0.02 seconds, synchronized with the execution of a new action.

## B DATASET DETAILS AND VISUALIZATION

### B.1 OVERVIEW

A total of 120 distinct villages were collected, comprising 6 types with 20 instances each. Our experiments were primarily conducted on this dataset of 120 villages. In addition, 18 different biomes and 8 types of locations were also gathered. Detailed information regarding the specific types of villages, biomes, and locations is provided in the table B.1 below.

Table 2: Minecraft locations: Villages, Biomes, and Structures

| Village | Biome | | Structure |
|---|---|---|---|
| Plains Village | Badlands | Plains | Woodland Mansion |
| Savanna Village | Beach | River | Ruined Portal |
| Snowy Village | Dark Forest | Savanna | Desert Temple |
| Taiga Village | Desert | Snowy Plains | Stronghold |
| Desert Village | Desert Hills | Snowy Mountains | Nether Bastion |
| Forest Village | Forest | Stony Shore | Nether Fortress |
| Zombie Village | Ice Spikes | Swamp | End Mainland |
| | Jungle | Taiga | End City |
| | Mushroom Fields | Mountains | |

For each location, we collected two types of trajectories: {**ABA, ABCA**}. For each type, trajectories were generated with four different navigation ranges: **5, 15, 30, and 50**. Each length comprises **20** distinct trajectories with varied start and end points.

This resulted in a total of 19,200 trajectories collected across all villages. With an average length of approximately 1,000 frames per trajectory, the dataset contains around **19.2 million** frames, equivalent to roughly **250 hours** of gameplay.

For experimental purposes, the villages were partitioned into training, validation, and test sets as follows: villages 1-16 were used for training, villages 17–18 for validation, and villages 19–20 for testing. The trajectories were not shuffled randomly across splits to prevent the model from memorizing structural features seen during training. Instead, this partitioning strategy encourages the model to reconstruct environmental structures from contextual information, as each village has a unique layout.

---

[6]https://github.com/PrismarineJS/prismarine-viewer

## B.2 DATA FORMAT

The smallest unit in our dataset consists of a pair of files: an **.avi video file** and a corresponding **.json file**, both of equal length. The .avi file stores the visual observations, while the .json file contains the associated state and action data. To avoid introducing inter-frame dependencies, we use MJPG (Motion JPEG) compression instead of H.264. Each frame is stored independently, and the resolution of the images is 640×360 at BGR format. The value of each BRG pixel is from [0,255]. The videos are recorded at 20 frames per second (FPS).

One step of recoreded trajectories in .json have following keys:

- x: current x coordinate, rounded to three decimal places.
- y: current y coordinate, namely hight, rounded to three decimal places.
- z: current z coordinate, rounded to three decimal places.
- yaw: the agent's horizontal viewing angle, measured in radians, ranging from $-\pi$ to $\pi$, where 0 indicates the agent is facing the positive z-axis.
- pitch: the agent's vertical viewing angle, measured in radians, ranging from $-\pi/2$ to $\pi/2$, where 0 indicates the agent is looking straight ahead (parallel to the ground plane).
- action: a dictionary with three possible keys: forward, jump, and camera. The values of forward and jump are booleans indicating whether the corresponding action is executed. The camera key holds a tuple [ yaw, pitch ], representing the **change** in the agent's viewing angle. Our sampling strategy ensures that these three actions do not occur simultaneously.
- goal: a dictionary containing the target's coordinates, with keys x and z representing the target position on the x- and z-axes, respectively.
- frame count: the index of the current frame. Due to rendering initialization, the first 20 frames are skipped, so frame count starts at 20. Since we ensure that the entire trajectory is recorded properly, the actual frame index within the trajectory can be obtained by simply subtracting 20.
- extra info: seed, location, navigation type $\in \{ABA, ABCA\}$, navigation range $\in \{5, 15, 30, 50\}$.

## B.3 STATISTICAL INFORMATION

### B.3.1 TRAJECTORY LENGTH

To better illustrate the scale and difficulty gradient of the dataset, the distribution of trajectory lengths is presented in the Figure 4 and Table B.3.1 below.

From the perspective of sequence length, even for the shortest navigation range (i.e., range = 5), the average trajectory length reaches as high as 180 frames. This implies that during evaluation, models must be capable of attending to visual inputs from over 100 frames ago. However, most current world models are limited to a history (context window) of only 32 frames, indicating that the required temporal context in our setting substantially exceeds the capacity of existing models.

In terms of distribution, different navigation ranges exhibit minimal overlap in their length distributions, which supports the validity of our difficulty curriculum design.

Table 3: Mean Trajectory Frames ($\pm$ Std) for Different Trajectory Types

| Trajectory Type | Range 5 | Range 15 | Range 30 | Range 50 |
|---|---|---|---|---|
| ABA | $180.5 \pm 30.4$ | $356.5 \pm 71.6$ | $627.1 \pm 130.3$ | $967.8 \pm 195.4$ |
| ABCA | $251.0 \pm 47.1$ | $544.7 \pm 113.1$ | $968.1 \pm 203.7$ | $1362.6 \pm 260.1$ |

### B.3.2 TRAJECTORY VARIANCE

Since both the $A \rightarrow B$ and $B \rightarrow A$ paths are generated using A* search, a natural question arises: Are the forward ($A \rightarrow B$) and backward ($B \rightarrow A$) paths different? If so, how significant is the discrepancy?

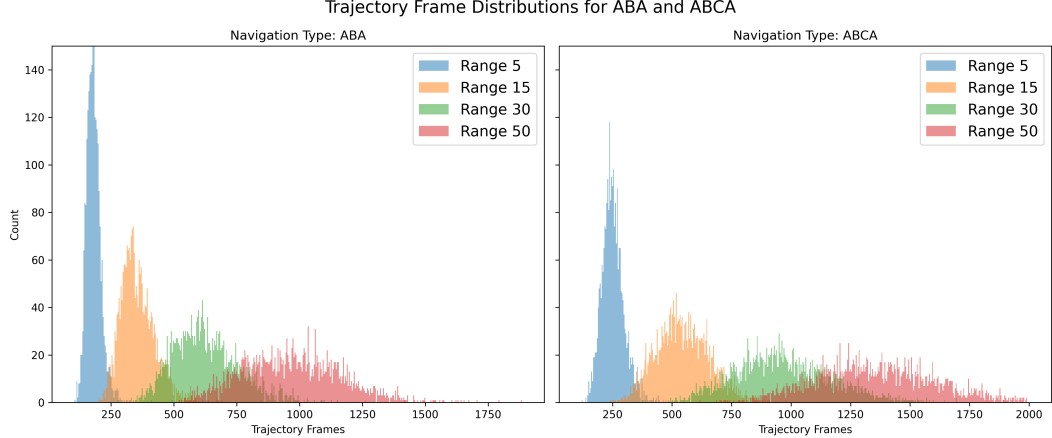

Figure 4: **Trajectory Frame Distribution of Navgation Type ABA and ABCA**

To quantify this difference, we compute the area enclosed by this loop to measure the spatial deviation between the two paths. Additionally, we normalize this area by the trajectory length to obtain a per-step deviation metric, which reflects the average divergence between forward and backward paths. The results are in the Table B.3.2.

Table 4: Deviation Between $A \to B$ and $B \to A$ Paths

| Metric | Range 5 | Range 15 | Range 30 | Range 50 |
|---|---|---|---|---|
| Enclosed Area | $1.83 \pm 2.20$ | $10.95 \pm 12.91$ | $39.20 \pm 45.20$ | $87.59 \pm 97.77$ |
| Normalized Deviation | $0.35 \pm 0.07$ | $0.67 \pm 0.13$ | $1.19 \pm 0.17$ | $1.62 \pm 0.20$ |

The results indicate that the $A \to B$ and $B \to A$ paths are not perfectly identical, but the deviation remains within an acceptable range. For the longest navigation distance of 50 grids, the average enclosed area between the two paths is 87.59, while the average total trajectory length is 108.39. If we approximate the path as forming a rectangle, this translates to about 1.62 grid units of deviation per unit path length on average. For shorter navigation distances (5, 15, 30 grids), the deviation is smaller.

Given that the navigation is driven by the A* algorithm, we believe this level of deviation is acceptable and does not significantly compromise spatial consistency or view reconstruction. Moreover, our bird's-eye visualizations that qualitatively illustrate forward and return paths.

### B.4 DATASET VISUALIZATION

We illustrate in Figure 5 two trajectories extracted from datasets with a navigation type of ABA and a navigation range of **5**. The upper subfigure corresponds to plains village (village ID:20, trajectory ID:05-10_14-09-44), while the lower subfigure is taken from snowy village (village ID:20, trajectory ID:05-10_17-46-35).

In the plains village, we observe that **frame 1 is visually similar to frame 129, and frame 41 is similar to frame 153.** In the snowy village, **frame 17 resembles frame 129, and frame 25 is similar to frame 153.** These observations suggest that, in order for a world model to accurately reconstruct future scenes, it must retain information from more than 100 frames earlier, and be capable of leveraging spatial context to infer and restore visual content.

To provide an intuitive visualization of the trajectories in our dataset, we present bird's-eye views of real trajectories. In Figure 6, the location is taken from plains village (village ID: 2). The left panel shows a trajectory of type ABA, while the right panel displays a trajectory of type ABCA. Yellow, blue, and green indicate navigation ranges of 5, 15, and 30, respectively. For visual clarity, trajectories with a navigation range of 50 are omitted.

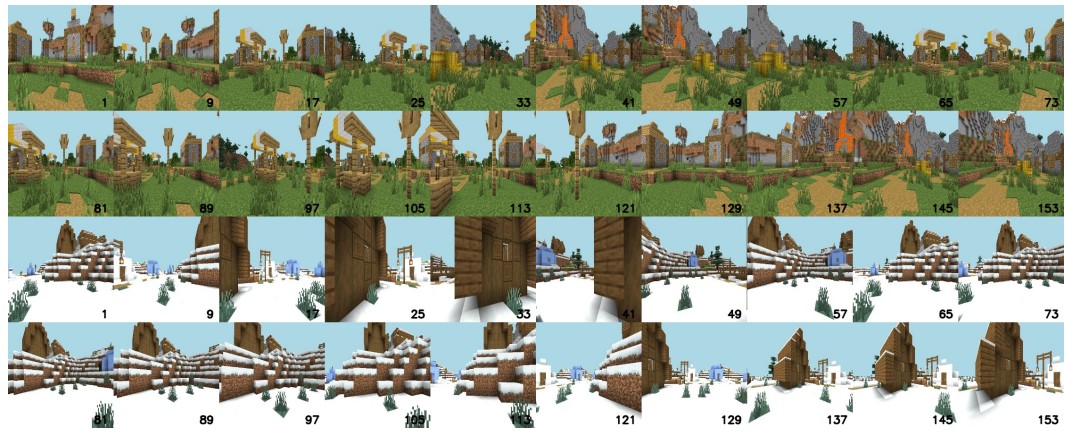

Figure 5: Demonstration of two LOOPNAV Trajectories. Upper half: Plains village. Lower Half: Snowy village. Black number indicates coresponding frame number.

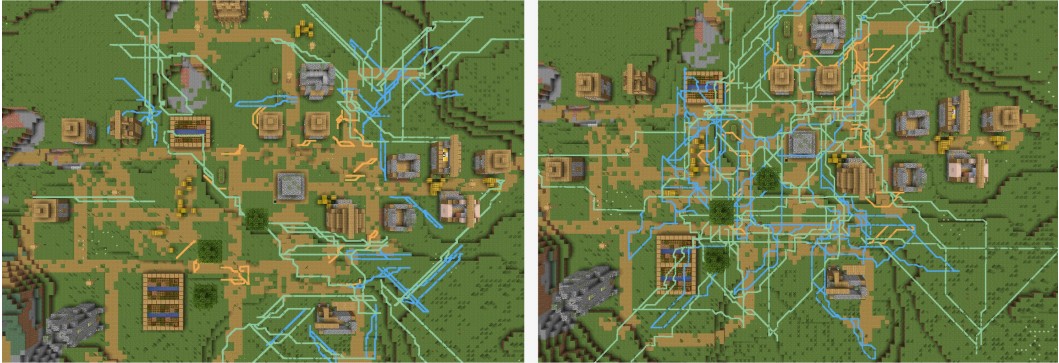

Figure 6: Demonstration of bird eye views for real dataset. Left: ABA type trajectories. Right: ABCA type trajectories. Yellow, blue, green indicate range 5, 15, 30 respectively.

## C EXPERIMENT SETTINGS AND RESULTS

We provide a detailed description of the experimental settings to facilitate reproduction. All inference and training experiments were conducted on NVIDIA GeForce RTX 4090 GPUs and A800-80GB, with a total compute time of approximately 1000 GPU hours respectively.

### C.1 OPEN OASIS

#### C.1.1 OASIS SETTINGS

We use the open-sourced Oasis-500M model for inference only. We do not train the Oasis model ourselves, as it is already pretrained on Minecraft VPT contractor data, and the official training code has not been released. We use 32 frames as the conditioning input, wthich is the maximum supported context length, and perform inference in an auto-regressive manner.

Their observation space is defined as (640, 360, 3), which is fully consistent with our dataset and thus requires no additional modifications. Their action space follows VPT's CameraQuantizer, with a maximum value of 20 and a bin size of 0.5, resulting in 40 discrete buckets that are normalized to the range [-1, 1]. In contrast, our actions are defined in radians within the range [-0.1, 0.1], and we directly use the raw values as actions. Additionally, their definitions of cameraX and cameraY are reversed compared to ours(and VPT's). Other actions, such as forward and jump, are consistent between the two settings. Inference hyperparameters are shown in Table 5

Table 5: Oasis Inference Hyperparameters

| Model | Video Encoder | Sampling |
|---|---|---|
| Model: `oasis-500M` | Model: ViT-VAE-L/20 | DDIM steps: 10 |
| Structure: DiT-S/2 | VAE patch size: 20 | Max noise level: 1000 |
| Model.max_frames: 32 | # Prompt frames: 32 | Noise absolute max: 20 |
| | | Stabilization level: 15 |
| | | Beta schedule: Sigmoid |

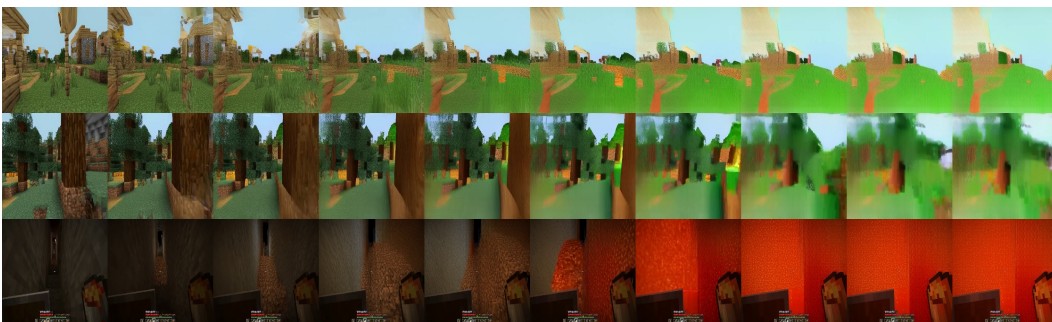

Figure 7: Results of Oasis. The first row corresponds to plains village, the second row to taiga village, and the third row shows the official inference demo. All three cases exhibit increasing blurriness.

### C.1.2 OASIS RESULTS

However, the results fall short of expectations. As shown in Figure 7, we observe that errors compound over time: small imperfections quickly snowball into collapsed frames. After approximately 30 frames, the generated images become increasingly blurred and fail to recover. We include a sample from plains village and a sample from taiga village. To demonstrate that the blurriness is not caused by our dataset or action configuration, we additionally include the official inference demo (Player729-f153ac423f61-20210806-224813, 256 frames) in line 3 for comparison.

### C.2 MINEWORLD

#### C.2.1 MINEWORLD SETTINGS

Mineworld is also a model pretrained on Minecraft VPT contractor data, and its official training code has not been released. For the same reason, we directly evaluate Mineworld on our benchmark without further training. Specifically, we evaluate the largest publicly available model, Mineworld-1200M-16f and Mineworld-1200M-32f, which support 16 frames and 32 frames as context respectively.

Mineworld supports up to 32 frames of total context, which includes both historical and previously generated frames. Its generation process is chunk-based. We experiment using sliding window strategies: using 31 frames as context to autoregressively generate the remaining frames.

Mineworld defines its observation space as 384×224. To align with this, we resize our original 640×360 frames to 384×224 using an area-based interpolation method, and apply standard normalization to the pixel values. In Mineworld, visual observations are further compressed by a VAE into a latent representation of size 24×14. As for the action space, since Mineworld uses degrees to represent camera rotations, we convert our radian-based actions into degrees before passing them to the model.

#### C.2.2 MINEWORLD RESULTS

To qualitatively evaluate the model's predictions, we visualize rollout results alongside ground truth frames in Figure 8. The first and third rows show rollouts for plains village (village ID:20, trajectory ID: 05-10_14-09-18) and desert village(village ID:20, trajectory ID: 05-10_14-11-32), respectively, while the second and fourth rows present their corresponding ground truth trajectories. We observe

that, unlike Oasis, Mineworld does not exhibit visual collapse. However, it similarly lacks spatial consistency and fails to reconstruct the corresponding scene structure.

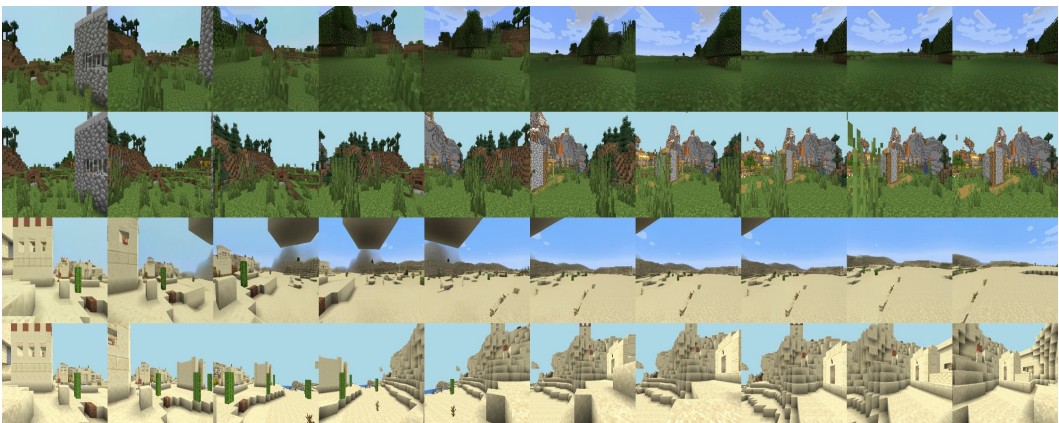

Figure 8: Results of Mineworld.The first and second rows show the rollout and ground truth for plains village, respectively; the third and fourth rows show the same for desert village.

### C.3 DIAMOND

#### C.3.1 DIAMOND SETTINGS

DIAMOND is a diffusion-based world model. We follow the `CS:GO` configuration and focus solely on training the world model, omitting the reinforcement learning (RL) agent training phase. For the observation space, following the default settings of the CS:GO branch in DIAMOND, we first resize the input images from $640 \times 360$ to $320 \times 180$, and then further downscale them to $64 \times 36$. Diffusion is performed on the $64 \times 36$ images, and an additional upsampler is trained to reconstruct the images back to $320 \times 180$.

For the action space, we adopt VPT-style camera quantization with a maximum value of 0.1 and a bin size of 0.02, resulting in 11 discrete bins for both yaw and pitch. Combined with forward and jump actions, the total action dimension is 24. The training hyperparameters are summarized in Table 6.

#### C.3.2 DIAMOND RESULTS

In Figure C.3.2, we present the rollout results of DIAMOND. We observe that DIAMOND does not suffer from visual collapse and is able to reconstruct the initial frames relatively accurately, which aligns with its context window of length 32. However, as the rollout progresses, the generated scenes gradually converge to empty grasslands, indicating that the model forgets previously observed structures. This suggests that DIAMOND, like others, fails to maintain spatial consistency over long horizons.

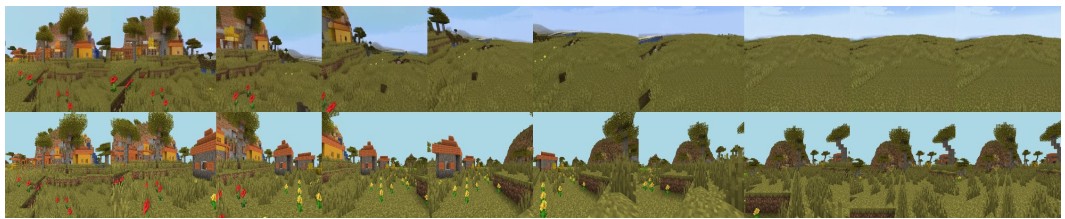

Figure 9: Results of Mineworld. First row: Rollout results. Second row: Ground truth.

Table 6: Model Configuration Parameters

| Category | Parameter | Denoiser | Upsampler |
|---|---|---|---|
| General | sigma_data | 0.5 | 0.5 |
| | sigma_offset_noise | 0.1 | 0.1 |
| | noise_previous_obs | true | false |
| | upsampling_factor | null | 5 |
| Inner Model | img_channels | 3 | 3 |
| | num_steps_conditioning | 4 | 1 |
| | cond_channels | 2048 | 2048 |
| Training | num_autoregressive_steps | 4 | 1 |
| | start_after_epochs | 0 | 0 |
| | steps_first_epoch | 400 | 400 |
| | steps_per_epoch | 400 | 400 |
| | sample_weights | null | null |
| | batch_size | 64 | 16 |
| | grad_acc_steps | 2 | 2 |
| | lr_warmup_steps | 100 | 100 |
| | max_grad_norm | 10.0 | 10.0 |
| Optimizer | lr | 1e-4 | 1e-4 |
| | weight_decay | 1e-2 | 1e-2 |
| | eps | 1e-8 | 1e-8 |
| Diffusion Sampler | num_steps_denoising | 3 | 10 |
| | sigma_min | 2e-3 | 1 |
| | sigma_max | 20.0 | 5.0 |
| | rho | 7 | 7 |
| | order | 1 | 1 |
| | s_churn | 0.0 | 10.0 |
| | s_tmin | 0.0 | 1 |
| | s_tmax | $\infty$ | 5 |
| | s_noise | 1.0 | 0.9 |
| | s_cond | 0.005 | 0 |

## C.4 NAVIGATION WORLD MODEL

### C.4.1 NWM SETTINGS

Navigation World Model (NWM) proposes a world model based on the Conditioned Diffusion Transformer (CDiT). Regarding the observation space, we first align with NWM's input format by decomposing trajectory videos into individual images. Following the official setting, each image is resized to 224×224, normalized, and then compressed into a 32×32 latent representation using the Stable Diffusion VAE.

For the action space, NWM differs from prior models in that it conditions on the agent's $(x, z)$ position and yaw angle, using absolute location and orientation to reconstruct future observations. Since our dataset records the $(x, z)$ coordinates and yaw at each step, we can directly adopt this format. Note that pitch information is omitted here. Empirically, we find that for navigation tasks, pitch tends to be less critical than yaw.

We train a CDiT-L/2 model with a context window of 4 frames. Detailed training hyperparameters are provided in Table C.4.1.

### C.4.2 NWM RESULTS

In Figure C.4.2, we present the rollout results of Navigation World Model. We observe that NWM retains a considerable amount of memory in the initial frames. However, the generated frames are noticeably blurry and distorted, suggesting that the model is unable to faithfully reconstruct spatial details or maintain long-range spatial consistency.

Table 7: NWM Training Configuration

| Parameter | Value |
|---|---|
| Batch size | 8 |
| Number of workers | 12 |
| Model | CDiT-L/2 |
| Learning rate | $8 \times 10^{-5}$ |
| Normalize action space | True |
| Gradient clipping value | 10.0 |
| Context size | 4 |
| *Distance Prediction* | |
| Min distance category | $-64$ |
| Max distance category | 64 |
| *Action Output* | |
| Predicted trajectory length | 64 |
| *Dataset Settings* | |
| Image size | 224 |
| Goals per observation | 4 |

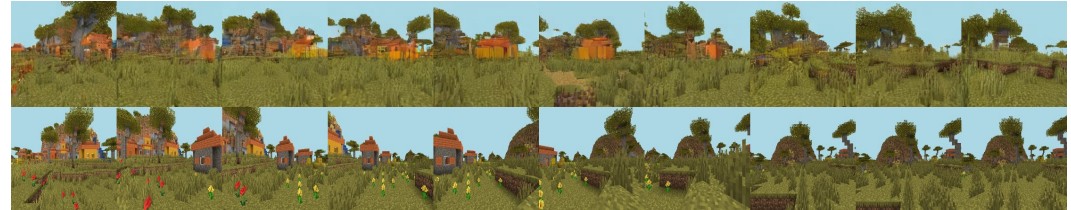

Figure 10: Results of NWM. First row: Rollout results. Second row: Ground truth.

