# OpenReview forum: "TOWARD MEMORY-AIDED WORLD MODELS: BENCHMARKING VIA SPATIAL CONSISTENCY"
_ICLR.cc/2026/Conference — Submitted to ICLR 2026_

### Official Review · Reviewer_xpSX · 2025-10-26

**Soundness:** 2
**Presentation:** 2
**Contribution:** 2
**Rating:** 4
**Confidence:** 4

**Summary:**

The paper presents LoopNav, a benchmark built in Minecraft to study spatial consistency in world models. It collects loop-style navigation videos so that models must reproduce the same scenes when revisiting locations. The benchmark measures how well models maintain spatial coherence over long sequences. The authors evaluate several existing world models using standard video metrics (FVD, LPIPS, SSIM) and show that current approaches still fail to keep consistent scene layouts over time.

**Strengths:**

- This paper investigates a timely and relevant research question given the popularity of video generation models.
- The proposed dataset seems to be quite large scale and appears to be a good test-bed for developing video generation models.

**Weaknesses:**

- Whether the A→B exploration context includes a 360° view at B? If not, the excessive amount of new observations during B→A would be highly unpredictable. Including these in the evaluation metrics could lead to biased results.

- Whether the A→B trajectory is guaranteed to be linear? If not, any intermediate point in the trajectory could be regarded as a point C, making A→B and A→B→C→A effectively equivalent?

- Whether solving LoopNav implies a world model that generalizes to other domains (e.g., unseen situations, other simulation environments, or the real world)? Some discussion or qualitative assessment of generalization would strengthen the paper.

- The benchmark cannot accommodate non-generative world models.

- There is no comparison with existing world-model benchmarks. Suggested related works include:
    - *3DLLM-Mem: Long-Term Spatial-Temporal Memory for Embodied 3D Large Language Models*
    - *World Consistency Score: A Unified Metric for Video Generation Quality*
    - *VBench: Comprehensive Benchmark Suite for Video Generative Models*
    - *WorldPrediction: A Benchmark for High-level World Modeling and Long-horizon Procedural Planning*
    - *GRASP: A Novel Benchmark for Evaluating Language Grounding and Situated Physics Understanding in Multimodal Language Models*
    - *IntPhys 2: Benchmarking Intuitive Physics Understanding in Complex Synthetic Environments*

**Questions:**

See Weaknesses

---

### Official Review · Reviewer_iwst · 2025-10-30

**Soundness:** 3
**Presentation:** 3
**Contribution:** 2
**Rating:** 4
**Confidence:** 5

**Summary:**

The paper presents LOOPNAV, a Minecraft dataset and benchmark to test long-horizon spatial consistency in action-conditioned world models using loop trajectories. On ~250 hours (~20M frames), return-leg evaluation with FVD/LPIPS/SSIM evaluation shows four existing models with short context windows drift over long rollouts, revealing weak memory.

**Strengths:**

1. The paper highlights spatial consistency, which is a very important capability and also challenge for world models. Improvements on such capability can large aid the application of world models in the field of embodied AI.
2. The paper contributes a large-scale dataset with diverse videos from the Minecraft platform.
3. The paper describes very clear on the data collection criteria and algorithms.

**Weaknesses:**

Major weaknesses:
1. The experiment and discussion is limited. There is only one main experiment in the paper and the discussion is not insightful. Many surprising results from the experiment is not fully analyzed and explained. The authors should use more dedicated experiments to answer the questions raised in the main experiment.
2. While the work underlines the importance of memory for world models, it does not come up with a proposal for any baseline solutions that use a memory module to improve existing video world models. This significantly weakens the contribution of this work.
3. The evaluation only contains 18 trajectories. This can lead to high bias during the evaluation, and a potential reason for the inconsistency in the experiment results.
4. The paper is very unclear on how the evaluation range is selected. What does the -5, -15, -30, -50 mean in Tab. 1? From Fig. 3, does it mean the context length is the same and the only difference is how the frames are downsampled temporally? If so, this design doesn't make much sense in terms of spatial memory.

Minor weaknesses:

5. The work focuses on the Minecraft platform. While the scene can be diverse, the environment is still in the grid-world patter and repetitive textures. The diversity is still not enough to evaluate general-purpose world models. Including data from other simulators like CARLA can significantly improve the diversity of the benchmark (I understand this can't be done during the rebuttal period.)
6. The qualitative result only includes data from one trajectory. It does not show a general performance patter of different models.

**Questions:**

I don't have questions for the paper. I like the motivation and ideas presented in the paper very much, and I'm happy to see a paper with such topic being accepted. Unfortunately, the current content in the paper does not meet the acceptance standard.

I will raise my rating to a positive one if all my major concerns are addressed during the rebuttal period.

---

### Official Review · Reviewer_UYtZ · 2025-11-01

**Soundness:** 2
**Presentation:** 3
**Contribution:** 3
**Rating:** 6
**Confidence:** 4

**Summary:**

This paper introduces a Minecraft-based benchmark designed to evaluate world models from the perspective of spatial consistency over long horizons, addressing the lack of suitable datasets in this area. The authors collected data from 150 distinct locations, totaling 50 hours with approximately 20 million frames, and evaluated four baseline models, including Oasis, Mineworld, DIAMOND, and NWM, across different navigation ranges.

**Strengths:**

Novelty and significance.

__S1__: This paper addresses the emerging challenge of the quadratic increase in computational and memory complexity found in most Transformer-based world models.

__S2__: The tool seems easy to use with large community support, making it useful in practice.

Quality of experiments.

__S3__: This paper conducted evaluations on four baseline models of Oasis, Mineworld, DIAMOND, and NWM over a spectrum of navigation ranges.

__S4__: It is great that discussions are provided for each experiment.

Clarity of presentation.

__S5__: The paper provides a reasonable level of detail and maintains decent presentation quality.

**Weaknesses:**

__W1__: __Lack of photorealistic environments for real-world scenarios.__ It is OK if the paper focuses on game-like simulation environment, but since the authors motivate the need of evaluating spatial consistency (Line 047 *“However, real-world exploration trajectories often span hundreds or thousands of frames...”*), it would be beneficial to better align the experiments with real-world settings.



__W2__: __Absence of results with explicit memory modules__. The paper would be more comprehensive if it included experiments with explicit memory components, which can be readily integrated into existing models. This omission contrasts with the authors’ stated motivation (Line 43: *“The memory module plays a critical role in addressing spatial inconsistency.”*).



__W3__: __The metrics used (FVD, LPIPS, SSIM) primarily assess visual fidelity__ rather than spatial consistency. While they may implicitly reflect spatial structure through ground-truth alignment, they fail to penalize cases where visually similar but semantically inconsistent structures (e.g., a road instead of a bridge) are generated. As the authors acknowledge on Line 91 *“Evaluation metrics often prioritize visual fidelity and short-term temporal smoothness over long-term spatial coherence or logical consistency”*, a dedicated metric explicitly designed to evaluate spatial consistency beyond the image space would significantly strengthen the benchmark.



__W4__:  __Lack spatial consistency and long-horizon metric__. It would strengthen the benchmark if a new metric that can capture both spatial consistency and long-horizon jointly is invented, although it may not be a requirement so I would not think this is a critical point to determine the score.



__W5__: __Intuition of Principle 1__. I find the intuition of Principle 1: *Visual Discriminability* somewhat unclear from a benchmark-design perspective. A new benchmark should ideally evaluate model capabilities that existing benchmarks and metrics fail to capture. However, manually enforcing visual discriminability, including ensuring frames are easily distinguishable over time, could inadvertently simplify the task for world models. This design choice may prevent the benchmark from evaluating a model’s true ability to maintain spatial consistency when frames are visually similar but spatially distinct, which is a more realistic and challenging scenario.



Therefore, I am not fully convinced by the value of Principle 1 in its current form. A more balanced approach that includes both visually discriminable and visually similar yet spatially complex trajectories could make the benchmark more meaningful and comprehensive.











__W6__: __A* vs Repeated Path__. I understand the rational of the use of the conventional A* algorithm for shortest-path planning when going back to the start point A. However, traversing the same path in reverse makes more sense to me as it has corresponding locations in both trajectories with various views, making it useful to evaluate world model’s spatial consistency. Please convince me if I am wrong.

__W7__: Although the dataset may seem to include diverse scenes shown in Figure 2, *“focus solely on village environments for training and evaluation”* in Sec. 5.2 may hinder the diversity of scenes represented in the empirical results.


---


__Minor Issues__ (Not Affecting Main Contributions) such as Typos, Grammar Mistakes, and Others:

__M1__: Text in Dataset Composition in Figure 2 is too small. It might not be visible in the printed paper.

__M2__: Line 011 “a crucial requirements” -> “a crucial requirement"

__M3__: Line 016-017 “However, there are no dataset designed..." -> “However, there are no datasets designed..."

__M4__: Line 105 “a explore-then-generate approach” -> “an explore-then-generate approach”

__M5__: Line 120 missed a space between “reinforcement learning“ and ‘(‘ in “reinforcement learning(Oh et al., 2015; Ha & Schmidhuber, 2018)”.

__M6__: Line 141 a redundant space is observed after “transformers”: “Recent approaches leverage transformers , and diffusion models”

__M7__: Line 410 “Unlike aformention models” -> “Unlike aformentioned models”.

__M8__: Line 442 missed a space between “increases” and ‘(‘ in “as the prediction horizon increases(Figure 3).”

---

Overall, this paper has a relatively high presentation quality, clarity, easy to follow and contains lots of details.

It attempts to tackle anrising important challenge to evaluate long-range spatial consistency, particularly as the complexity of most Transformer models grows quadratically in both computation and memory with respect to context length.


However, there remain some fundamental issues in the benchmark’s design principles, such as the lack of structure-specific metrics and the absence of evaluations involving memory-module models. These limitations prevent it from being an excellent benchmark, as detailed in the weaknesses. Given its high presentation quality, I assign a score of 6, placing it near the borderline.

**Questions:**

__Q1__: __Unclear “Context” meaning in Table1__. It took me a while to  “guess” it refers to *“Most models operate with a context length of 32 frames,”* from Line 436. It might be better to explicitly explain it as context length in the caption.



__Q2__: __Unclear “spatial consistency” definition__. The paper does not clearly define spatial consistency. It is unclear whether this refers to (a) the spatial coherence of trajectories in the map or latent space, (b) the preservation of visual spatial structures in the image space, or (c) consistent reconstruction over time. Although Line 35 defines it as “*the ability to preserve coherent and stable spatial structures across time,*” the formulation remains ambiguous and would benefit from a more concrete, operational definition. For readers familiar with prior work, (b) seems to be the intended meaning. Nonetheless, making this explicit would reduce ambiguity.





__Q3__: I actually don’t quite understand the fundamental differences between ABA and ABCA as they all represent “Go to somewhere else (B or BC) and back to the same place”. They look the same as A\*A where * denotes “somewhere else” . The only difference is that * here can be scaled to longer trajectories such as B -> BC. To me * can be scaled, so ABA, ABCA, ABCDA or ABCD...A can be represented by A\*A.

---

### Official Review · Reviewer_SxSs · 2025-11-01

**Soundness:** 2
**Presentation:** 3
**Contribution:** 2
**Rating:** 4
**Confidence:** 3

**Summary:**

The paper focuses on evaluating the spatial consistency capability of world models, particularly how well they retain memory of historical observations during long-horizon navigation. To address the lack of suitable benchmarks, the authors propose a new Minecraft-based dataset, which records loop trajectories in two forms and provides challenging environments with repeated visits. With this setup, the benchmark is able to test the memory robustness and consistency of world models. The authors evaluate four different representative world models and show that all of them struggle on the proposed tasks, revealing the difficulty of maintaining long-term spatial coherence.

**Strengths:**

* The motivation to evaluate memory consistency is well-founded and represents an important research direction.


* The experiments effectively highlight the challenges of maintaining spatial consistency in existing methods.


* The action-conditioned data generation pipeline in Minecraft is well-designed and shows potential for scalability in both training and testing scenarios.

**Weaknesses:**

* Although the experiments highlight the challenges of this benchmark, there are no experiments comparing the performance of the same model with and without training on the dataset.
* The model’s spatial consistency ability may change over time; additional results analyzing performance across different time horizons would be helpful.
* The generated trajectories are still difficult to demonstrate as matching the real distribution, which may limit the benchmark’s overall value. Moreover, given that recording navigation data is relatively easy in simulators like Habitat, the choice of Minecraft as the testing environment is not fully justified

**Questions:**

What are the advantages of using a Minecraft environment compared to a standard navigation simulator?

---

### Meta-Review · Area_Chair_fqMH · 2026-01-06

**Summary:**

All four reviewers acknowledged the paper' motivation to evaluate spatial consistency in world models through loop-based navigation trajectories in Minecraft. However, they converged on several fundamental concerns. The primary weaknesses identified include the misalignment between motivation and execution, with reviewers noting the paper emphasizes memory modules yet provides no experiments with explicit memory components (R_UYt, R_iwst). The evaluation metrics (FVD, LPIPS, SSIM) measure visual fidelity rather than spatial consistency directly (R_UYtZ), and no dedicated spatial consistency metric was developed. The experimental evaluation was deemed limited, with only 18 trajectories tested (R_iwst), unclear evaluation range definitions (R_iwst), and no ablation studies comparing model performance with and without training on the proposed dataset (R_SxSs). Additional concerns included the lack of photorealistic environments despite real-world motivation (R_UYt), insufficient justification for choosing Minecraft over standard navigation simulators (R_SxSs), limited scene diversity by focusing only on village environments (R_UYt), and missing comparisons with existing world model benchmarks (R_xpSX). Design principle concerns were also raised, particularly regarding Principle 1's visual discriminability potentially oversimplifying the task (R_UYt).

**Reviewer Concerns:**

There was no rebuttal submitted by the authors. The absence of a rebuttal meant that none of the substantial concerns raised by reviewers were addressed, including fundamental questions about evaluation design, metric appropriateness, experimental comprehensiveness, and the missing memory module experiments that align with the paper's core motivation.

**Reviewer Scores:**

Given the lack of rebuttal and discussion, reviewer scores would likely have remained unchanged or potentially decreased.

---

### Decision · Program_Chairs · 2026-01-26

Reject